

# IoT-based 4-dimensional high-density electrical instrument for geophysical prospecting

Keyu Zhou[1], Qisheng Zhang[1], Yongdong Liu[2], Zhen Wu[1], Zucan Lin[1], Bentian Zhao[1], Xingyuan Jiang[1], Pengyu Li[1]

[1] School of Geophysics and Information Technology, China University of Geosciences (Beijing), Beijing, China
[2] Beijing Geoscience Exploration Technology Co., Ltd, Beijing, China

*Correspondence to*: Qisheng Zhang(zqs@cugb.edu.cn)

**Abstract.** The high-density electrical method is a primary method used in shallow geophysical prospecting. With the rapid industrial development of recent years, the function and performance of high-density electrical instruments have been considerably improved in several aspects. However, most of the electrical instruments currently available on the market still exhibit some shortcomings, such as being bulky, heavy, limited in their data acquisition accuracy, and difficult to connect to the Internet for remote monitoring. To address these problems, this study developed a new multifunctional 4-dimensional (4D) high-density electrical instrument based on remote wireless communication technology. The system is small and lightweight, includes an integrated transceiver, has high data acquisition accuracy, and is capable of remote wireless real-time control. In this study, the hardware circuit was designed. The Arm all-in-one (AIO) LJD-eWinV5-ST7 with a 154.4cm × 87cm, 800 × 480 high-brightness wide-temperature-range display is used as the host computer, which has the advantages of small size, low power consumption, and abundant hardware resources. IoT technology is incorporated in the system, and a 4G module is employed to provide a real-time remote control and data acquisition monitoring system based on the cloud platform. Tests showed that this instrument is stable and convenient to use and can meet the requirements for use in field prospecting.

## 1 Introduction

Energy serves as the foundation for social and economic development, and geological work serves all aspects of the social economy. With the rapid development of science and technology and the fast growth of national economies, there is an increasing demand for energy in all countries. It is pointed out in the *2019 China Mineral Resources (CMR) Report* (Ministry of natural resources, 2019) that China's consumption of primary energy, crude steel, gold, 10 non-ferrous metals, and cement remains the highest in the world. Therefore, continuing endeavors to tap and develop the potential of domestic resources and increase research efforts as well as improve resource supply, and guarantee capabilities are still important measures for ensuring ongoing economic development.

There are several challenges that arise in the process of resource extraction. It is necessary to assess the geological conditions of the location targeted for extraction. Different rocks and minerals possess different physical properties. In geophysical





prospecting, these differences are observed and studied, allowing mineral resources to be assessed indirectly through an exploration of the stratum lithology and geological structure. Relative to other prospecting methods such as drilling, this method offers the advantages of high efficiency, high resolution, strong imaging effectiveness, low cost, and low destructiveness, among others. Electrical prospecting is an important branch of geophysical prospecting methods. It observes and analyzes the spatial and temporal distributions and propagation of the earth's electric field based on the different electrical

properties of the underground rocks and minerals, thereby identifying the geological structures in the stratum and solving the corresponding geological problems. This type of geophysical prospecting explores and searches for distributions of underground mineral deposits having extraction value (Zhang et al., 2013). Among its advantages are high resolution, accurate results, and ease of use. However, as conventional electrical prospecting can obtain only one set of data from each power supply measurement, the amount of information acquired is small, and the speed of measurement is slow. As a result, the

geological information it provides on the structural characteristics of the geoelectric section is sparse and therefore not readily processed statistically. Thus, the concept of a high-density electrical method has been proposed. The high-density electrical prospecting technique arranges hundreds of electrodes at a time and uses an electrode multiplexer to switch between them to direct the power and control the measurements. In addition to indicating the lithological changes of the prospected geological body horizontally at a given depth, this method can represent differences that run vertically. The high-density electrical

instrument can obtain highly accurate and rich geological information and thereby play an important role in prospecting for minerals, water, specific hydrogeological conditions, underground buildings, caves, and geofractures (Yang and Tian, 2014; Wang et al., 2013; Gu et al., 2010; Yang, 2011). With improvements in the depth of detection, observation accuracy, and the diversity of observation forms, the range of applications of high-density electrical methods has also been expanding (Zhou et al., 2018; Song et al., 2020), offering broad promise for further development. Development of these methods will continue to

trend toward multiple channels, multiple parameters, multiple dimensions, multiple functions, high power, and wide ranges (Yan et al., 2012).

Owing to the complexity of conditions on the surface and underground, electrical instruments used in prospecting complex terrain continuously face new requirements and challenges and must be constantly improved. At present, most electrical instruments available on the market are bulky and heavy, limited in their data acquisition accuracy, and difficult to connect to

the Internet for remote monitoring. Therefore, it is important to continuously seek ways to improve upon the conventional high-density electrical instruments.

## 2 Conceptual principles

Suppose that power is supplied to the underground via two power supply electrodes A and B on the horizontal ground surface as shown in Fig. 1. The current is $I$, and the underground half-space – which is a uniform, infinite, and isotropic medium – has

a resistance of $\rho$. Then the difference in potential between any two measurement electrodes M and N on the ground's surface can be expressed as follows:


$$\Delta U_{\text{MN}} = \frac{\rho_I}{2\pi}\left(\frac{1}{AM} - \frac{1}{BM} - \frac{1}{AN} + \frac{1}{BN}\right).$$ (1)

Thus, earth resistivity can be obtained as follows:

$$\rho = \frac{2\pi}{\frac{1}{AM} - \frac{1}{BM} - \frac{1}{AN} + \frac{1}{BN}} \frac{\Delta U_{\text{MN}}}{I}.$$ (2)


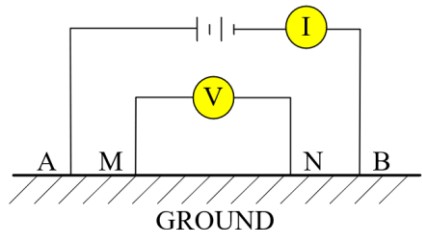

**Figure 1: Principle of the basic electrical method.**

The theory underlying the high-density resistivity method is the same as in the conventional resistivity method; the difference is that the former has a higher-density arrangement of measurement points for taking observations. During the measurement
process (Fig. 2), once all the electrodes are placed on the measurement points at the specified interval, the host can automatically control the changes in the power supply electrodes and the receiving electrodes, thereby completing the measurement (Dong and Wang, 2003). In terms of design and technical implementation, the high-density electrical measurement system is based on advanced automatic control theory and uses large-scale integrated circuits. A large number of electrodes are used, which can be combined freely. In this way, more geoelectric information can be extracted, and the
multi-coverage observation approach used in seismic prospecting can be realized in electrical prospecting as well (Di et al., 2003).

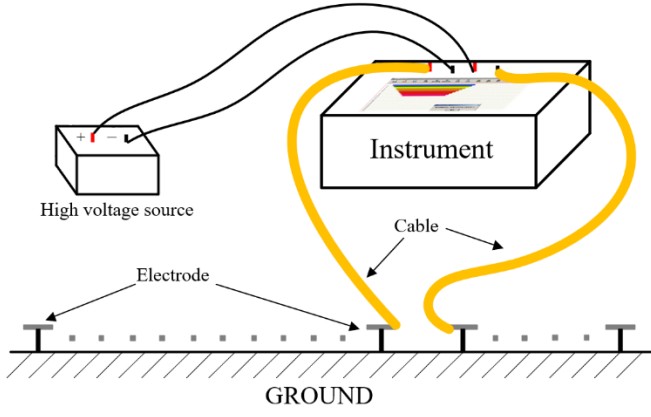

**Figure 2: Principle of the high-density electrical method.**



## 3 Design of 4-dimensional (4D) high-density electrical instrument based on IoT

### 3.1 Overview of architecture

Development of the system used in this study included designing hardware circuits, software programs, and mechanical structures. The system's stability and accuracy in actual use are particularly important (Mendecki et al., 2010).

A block diagram of the IoT-based 4D high-density electrical instrument is shown in Fig. 3. The instrument consists mainly of a high-voltage board, an acquisition board (AB), and an Arm all-in-one (AIO) PC, which form a complete system with the electrodes and electrode multiplexers. The AIO is the Arm LJD-eWinV5-ST7 with a 154.4cm × 87cm, 800 × 480 high-brightness wide-temperature-range display. It is small in size, consumes little power, and has abundant hardware resources, including Secure Digital (SD) cards and multiple RS232 and RS485 ports. The system additionally contains a 4G module, which uses the USR-G780 module. This enables the user to not only control the instrument at the measurement site directly via the mouse or touch screen but also control it remotely with a computer via the cloud platform. To carry out the 4D high-density electrical prospecting, 480 electrodes are available for use.

The system design is described in detail in the following subsections.





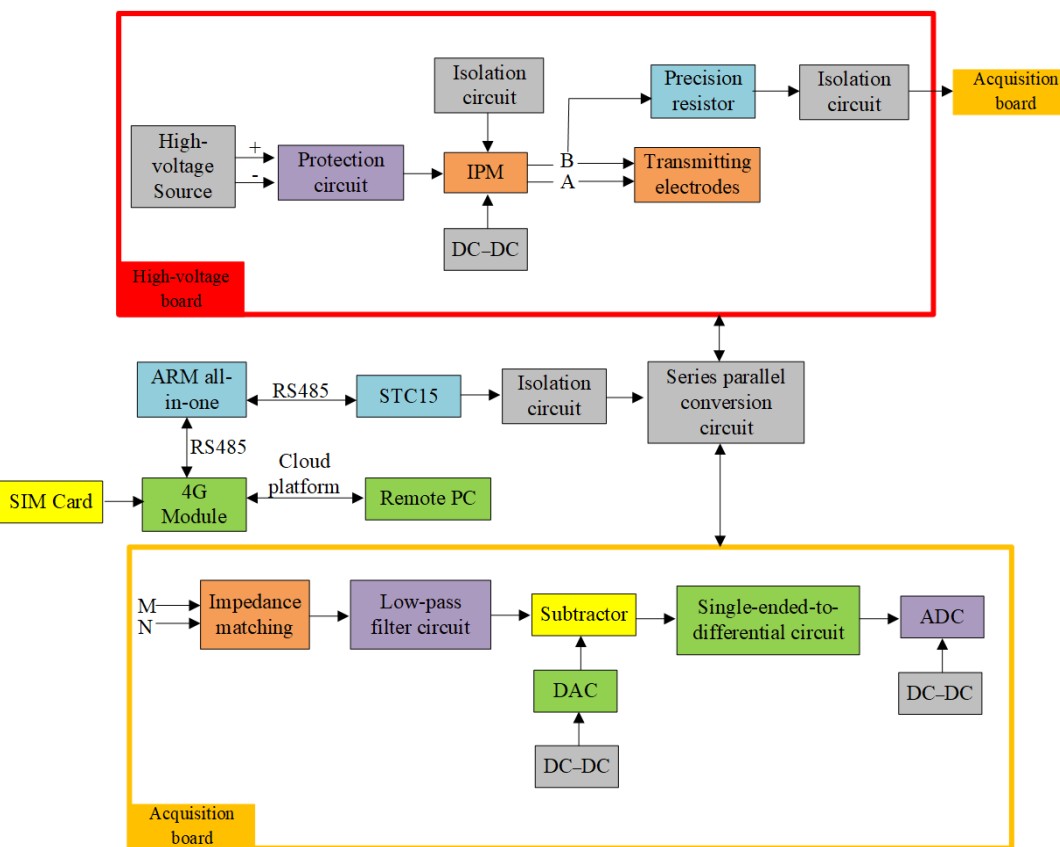

**Figure 3: High-level architecture of the 4D high-density electrical instrument based on IoT.**

## 3.2 Design of acquisition circuit

The AB serves as the front-end circuit, which is responsible for processing the analog signals acquired by the electrodes and converting them into digital signals to be processed. The ADC chip is a high-precision 24-bit serial A/D converter CS5532. The chip's gain can be controlled, and it contains a very-low-noise, chopper-stabilized instrumentation amplifier, and a fourth-order delta–sigma modulator followed by a digital filter.

The front ends of the ADC chip's two channels (ch1 and ch2) are respectively connected to the precision resistors of the MN and AB electrodes' front ends. The voltage and current values can be read simultaneously using dual channels. Figure 4 shows a block diagram of the acquisition device. The processing of the two analog channels is similar, the difference being that the MN voltage channel has self-potential compensation. Figure 5 shows a schematic of the MN channel of the acquisition device.

The signals first undergo impedance matching and then pass through the low-pass filter to filter out high-frequency clutter. Because the input to the ADC is a differential signal, the single-ended signal must be converted into a differential signal to enter the ADC chip, which then outputs the digital signal. The ADC chip of this circuit performs sampling at 20 ms (50 Hz corresponds to a 20 ms period), thereby effectively suppressing the interference from the 50 Hz grid power and multiple waves.





In addition, the channel connecting AD and MN has an INA128 as a subtractor, which subtracts the natural self-potential ($V_{SP}$)

converted by the DAC chip, thereby accomplishing self-potential compensation, for a more accurate calculation result.

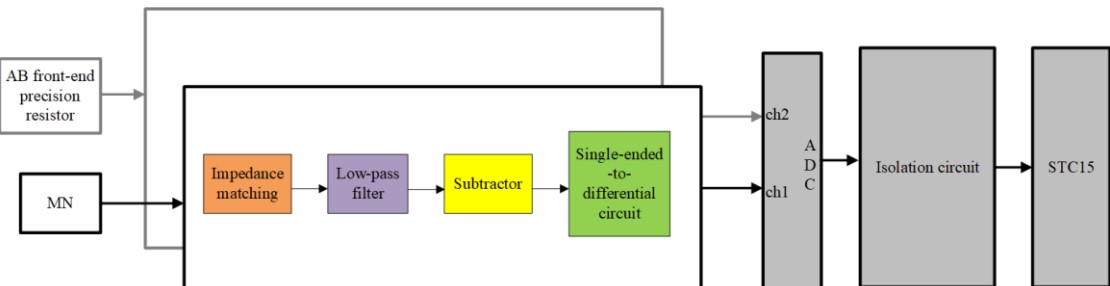

**Figure 4: Block diagram of the AB.**

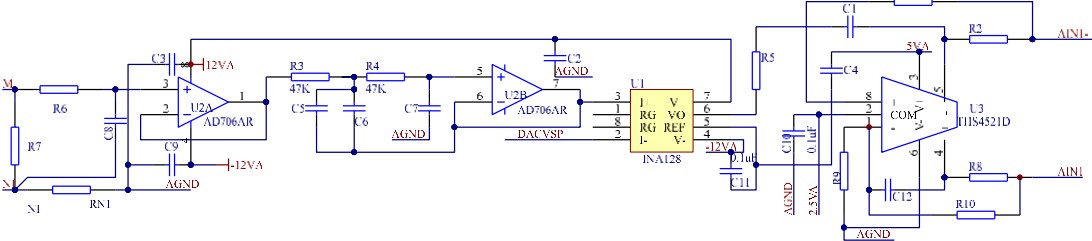


**Figure 5: Schematic of the MN channel on the AB.**

### 3.3 Design of transmitting circuit

The transmitting circuit can be divided into alternating current (AC) transmission and direct current (DC) transmission, and different devices are applied in different environments and for different measurements (Li et al., 2020; Zeng et al., 2018).

This system uses a DC power supply. Compared with an AC power supply, DC provides a greater prospecting depth and exhibits advantages in reducing electromagnetic coupling interference. The transmitting circuit consists mainly of a protection circuit, an isolation circuit, the intelligent power module (IPM) (10 A / 1200 V) (Xing, 2001), a power conversion circuit, the output electrodes, and a high-voltage source. The external DC high-voltage power supply is connected to the IPM via the protection circuit. The main control unit sends control commands through the multiple optocoupler to control the IPM. The

optocoupler used here is the HCPL-4504 from Agilent, which is an optocoupler dedicated to the IPM interface. This product's emitting light source and receiving light source are insulated by a transparent insulator, and there is no actual electrical connection. It has high speed, a high common-mode ratio, and an extremely short parasitic delay. Its instantaneous common-mode ratio is greater than 10 kV μs$^{-1}$. It is an electrical isolation device dedicated to the IPM, which effectively reduces mutual interference between signals.


This system is an integrated transceiver having no separate transmitter; it exhibits the advantages of being lightweight, small, and portable.

### 3.4 Software design

The Arm AIO LJD-eWinV5-ST7 has abundant hardware resources and a complete set of software functions. It contains, for example, an I/O port, SD card, dual-channel ADC, multiple RS232 ports, and multiple RS485 ports. The built-in ADC

sampling rate is fixed at 2 µs, which is too fast and produces too much noise. Therefore, an external ADC chip is required. Then, because the I/O port's timing accuracy cannot meet the requirement of this external ADC chip, an STC15 single-chip microcomputer is added as a slave computer to control commands. It communicates with the Arm AIO through an RS485 interface. Therefore, the software system includes the STC15 software and the Arm AIO software.

The slave computer program is the STC15 microcontroller program based on the Keil4 platform and is used to perform current

and voltage acquisition, self-potential compensation, data transmission, communication, and other functions. It also sends the acquired current and voltage values to the Arm AIO for calculation and processing.

C# is an object-oriented programming language derived from C and C++. It inherits the powerful functions of both C and C++ while excluding some of their complex features (Perkins et al., 2016; Zhao et al., 2013) and combines the simple visual operation of VB with the operating efficiency of C++ (Ji, 2008). This study used the C# language, a control platform based on

the Windows CE operating system, and the VS2008 development software. The software functions include creating new projects, setting parameters (device type, measurement mode, electrode distance, power supply time, power-off time, starting electrode, ending electrode, starting measurement layer, ending measurement layer, power supply current, secondary delay, use/nonuse of $V_{SP}$ compensation, etc.), opening a project, assigning addresses, inspecting electrodes, executing measurement commands, monitoring commands, viewing profiles, viewing curves, executing the END command, saving files, viewing data,

pausing, and stopping. The various device types include the Wenner device, Wenner–Schlumberger device, Schlumberger device, β device, Wenner roll-along, pole–pole device, pole–dipole device, differential device, and dipole–dipole device. An overview of the instrument's program flow is given in Fig. 6.





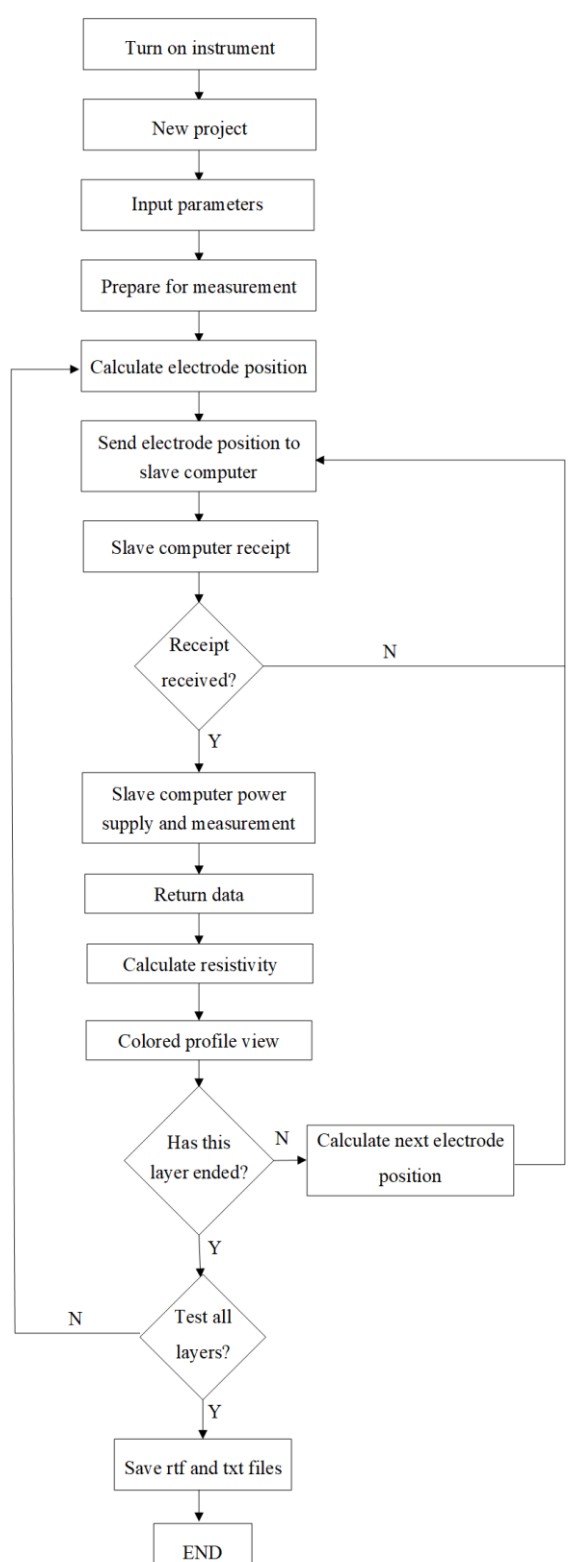



**Figure 6: Flowchart of the instrument's program.**

**3.5 4G module**

The use of sophisticated Narrowband IoT (NB-IoT) technologies enables long-distance wireless communication between a control center and a high-density electrical instrument (Li et al., 2019). This study used the USR-G780 module, which is an M2M product launched in 2016. It supports 4G high-speed access for China Mobile, China Unicom, and China Telecom as well as 3G and 2G access for China Mobile and China Unicom. The software includes a complete set of functions, covering the majority of conventional application scenarios. Users can perform transparent two-way data transmission from the serial port to the network by using simple settings. It also supports the custom registration package, heartbeat package function, four-way socket connection, and transparent cloud access. The USR-G780 was developed using an embedded Linux system and has high stability. Under the 4G network, it exhibits high speed and low latency and is therefore suitable in scenarios involving the transmission of large quantities of data and frequent interactions (Chen et al., 2015; Wang, 2017; Wang et al., 2017; Jia et al., 2016). Thus, it was an appropriate choice for the needs of this study. The connection topology is shown in Fig. 7.

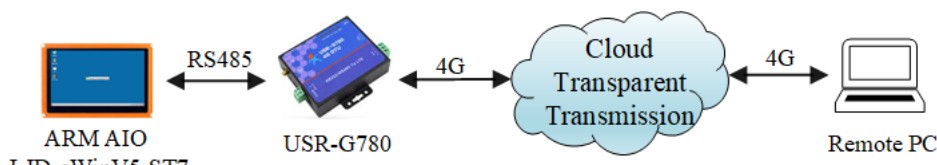

**Figure 7: Connection topology.**

**3.6 Key performance indicators of the instrument and a comparison**

The 4D high-density electrical method adds a time dimension to the 3-dimensional (3D) method; that is, $\rho = \rho(x,y,z,t)$. The data acquisition is performed by arranging the same set of electrodes at a given location and then repeating the 3D data acquisition at different time points (Loke, 1999). Table 1 lists representative performance indicators of this electrical instrument and, for comparison, those of the widely used Geopen E60 series and the DUK-2A high-density electrical 175 instrument of Chongqing Geological Instrument Co., Ltd.



**Table 1: Comparison of instrument performance indicators**

| Item | High-density electrical instrument of this study | Geopen E60 series high-density electrical instrument | DUK-2A high-density electrical instrument of Chongqing Geological Instrument Co., Ltd. |
|---|---|---|---|
| **Measurement modes** | Electrical resistivity, polarizability, natural potential | Electrical resistivity, polarizability, natural potential | Electrical resistivity, polarizability, natural potential |
| **Functions** | Transmitter, receiver | Transmitter, receiver | Transmitter, receiver |
| **Maximum output current (A)** | 5 | 1.5 | 5 |
| **Type** | Distributed | Distributed | Centralized |
| **Power (W)** | 5000 | 400 | 4500 |
| **Measure button location** | Integrated main keyboard + PC remote control | Integrated main keyboard | Integrated main keyboard |
| **Dimensions (mm)** | $320 \times 210 \times 110$ | $360 \times 230 \times 100$ | $305 \times 200 \times 202$ |
| **Weight (kg)** | 4 | 6 | 8 |

## 4 Instrument tests

### 4.1 Laboratory and campus tests

After the instrument was developed, its practical application capabilities needed to be tested. To test the instrument's accuracy in the laboratory, an experiment was conducted using the resistor string method. A resistor string network was soldered as shown in Fig. 8 to form the equivalent of a dummy load. There were forty-eight 510 Ω resistors, with 1 Ω resistors placed between adjacent 510 Ω resistors, forming a voltage divider network. From each 510 Ω resistor, a wire was soldered from the

board to the cable, forming the equivalent of 48 measurement electrodes. The voltage and current data measured at a 60 V power supply were recorded, with results as shown in Fig. 9. After several experimental runs, including tests with large and small signals (range from 1mV to 500V ) and a variety of resistance values (range from 0.01 Ω to 510 Ω), the final accuracy was calculated to be within 1/1000.



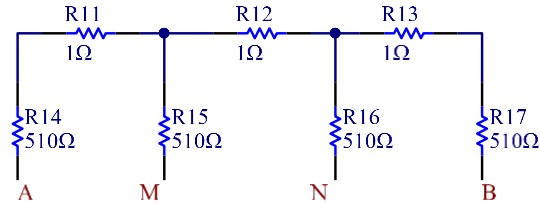


**Figure 8: Schematic of resistor string connections for laboratory experiment.**

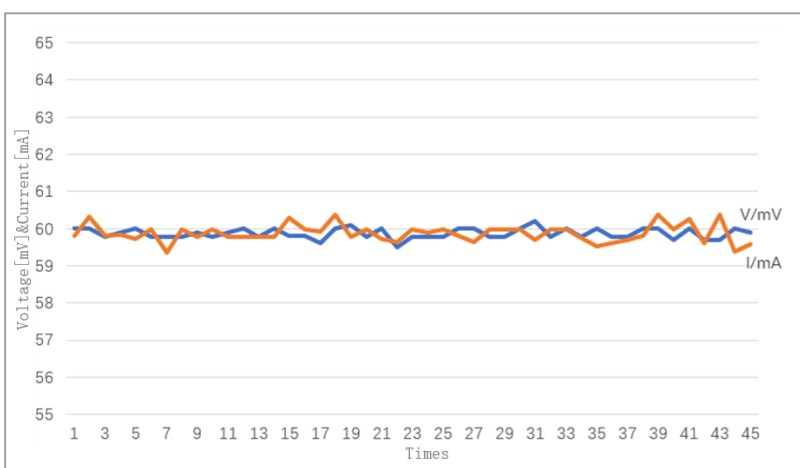

**Figure 9: Data results from laboratory experiment.**

Further testing was conducted on the China University of Geosciences (Beijing) campus, as shown in the Fig. 10 photo. A large lawn in front of the main building was selected as the ground to be tested; there is a tube well on the lawn. The electrodes were arranged in a space of approximately 60 m. A real-time profile view of the test site as displayed on the instrument is shown in Fig. 11; it can be seen that the lawn and the tube well have different colors on the profile view. To validate the remote

solution, in addition to adjusting the 4G module in the laboratory, transparent cloud transmission was used to monitor the data remotely on the PC in the laboratory. The profile view displayed on the notebook PC (Fig. 12) was the same as that displayed on the electrical instrument. Thus, the feasibility of using the program remotely was confirmed.





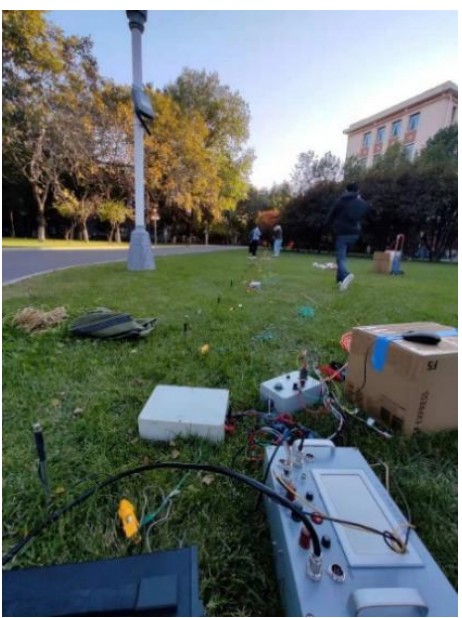

**Figure 10: Photo of the scene of the campus test.**

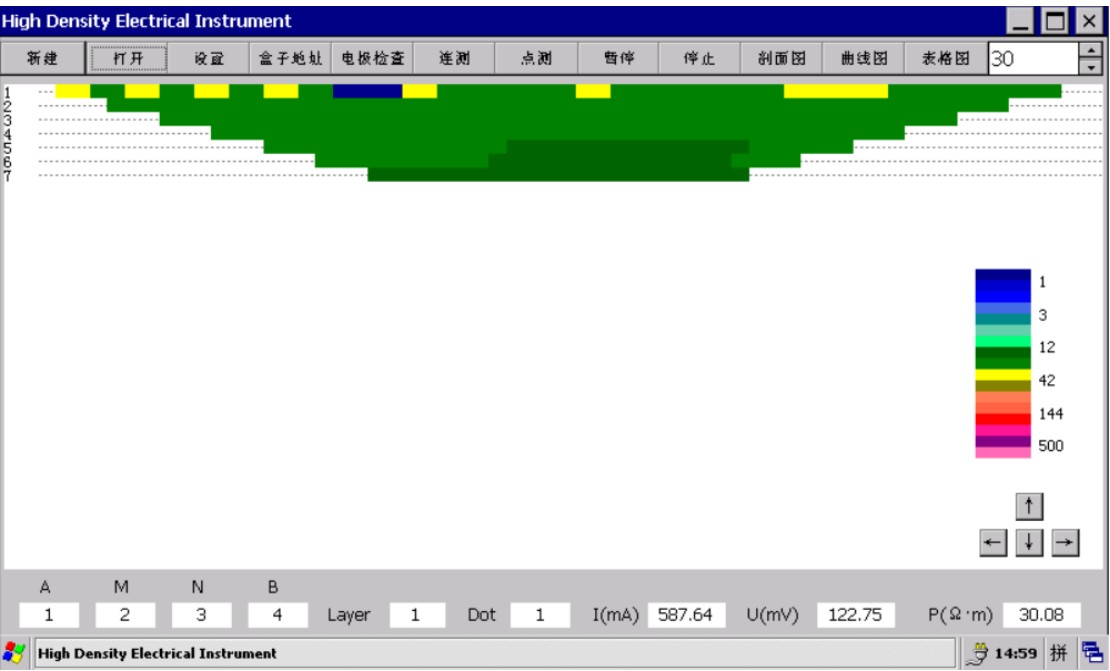

**Figure 11: Real-time profile view of the test site as displayed on the electrical instrument.**






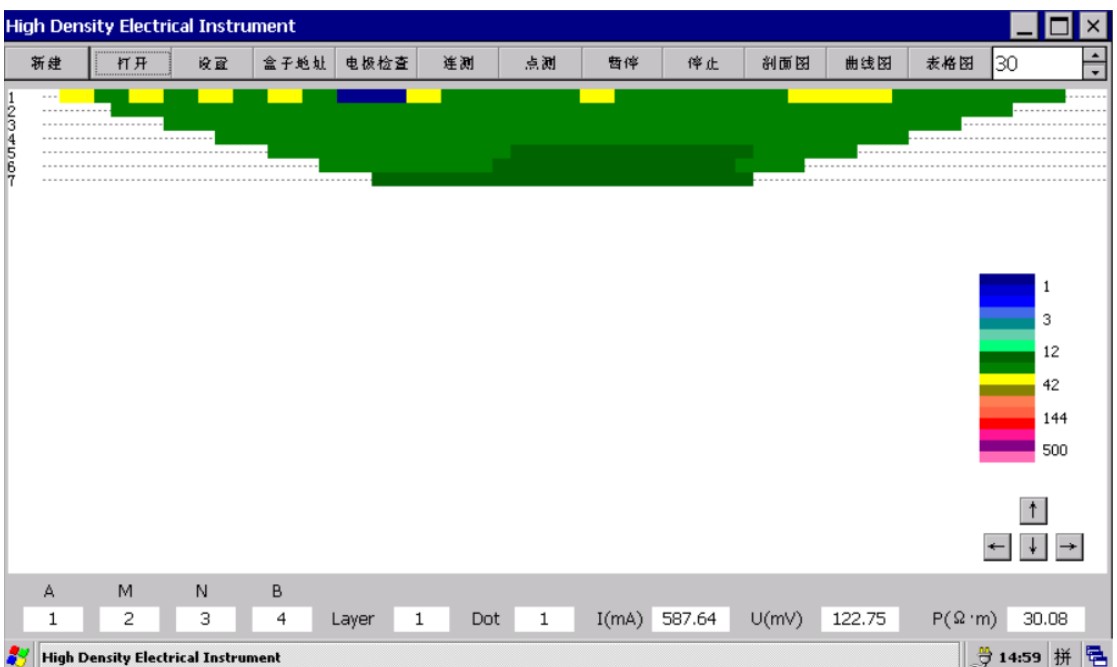

**Figure 12: Real-time profile view of the test site as displayed on the remote PC.**

**4.2 Field tests**

To further verify the stability and feasibility of the instrument, it was used to conduct high-density electrical prospecting in Lantian Village, Anxi County, in the province of Fujian, China. In correspondence with the actual terrain and topography of

the farmland on the east side of Huazhang Natural Village, Lantian Village, Lantian Township, three high-density electrical survey lines were arranged (Fig. 13 shows their schematic layout), with electrodes spaced at 10 m on each survey line. Survey Line L1 had 96 electrodes, whose starting and ending coordinates were (25.132667° N, 117.879972° E) and (25.124778° N, 117.878306° E), respectively; L2 had 64 electrodes, whose starting and ending coordinates were (25.124000° N, 117.883417° E) and (25.126611° N, 117.878528° E), respectively; and L3 had 80 electrodes, whose starting and ending coordinates were

(25.132250° N, 117.880250° E) and (25.125889° N, 117.878750° E), respectively. A Garmin Vista handheld GPS was used to position each electrode. Figure 14 shows a photo of the field testing.



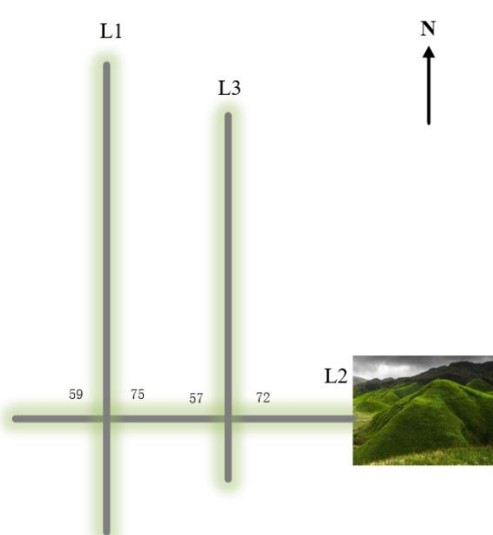

**Figure 13: Schematic layout of the survey lines. (The 75th measurement point on the L1 survey line intersects with the 59th measurement point on the L2 survey line, and the 72nd measurement point on the L3 survey line intersects with the 57th measurement point on the L2 survey line.)**

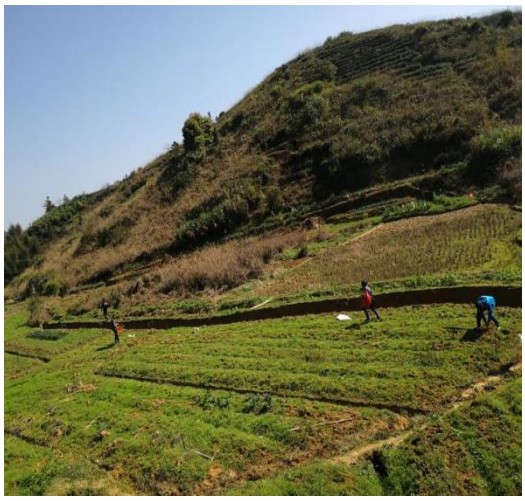

**Figure 14: Photo of the field testing.**

### 4.2.1 Survey Line L1

Survey Line L1 had 96 electrodes, a trace spacing of 10 m, and a profile length of 960 m. The latitude, longitude, and elevation of the first electrode were 25.132667° N, 117.879972° E, and 612 m, respectively. Taking the 20th electrode measurement point (with latitude, longitude, and elevation of 25.131028° N, 117.879944° E, and 611 m, respectively) as the center, the zone



beneath exhibited low resistance, and the greatest depth was approximately 70 m. From the 38th electrode measurement point

(with latitude, longitude, and elevation of 25.129472° N, 117.880056° E, and 614 m, respectively) to the 56th electrode

measurement point (with latitude, longitude, and elevation of 25.128000° N, 117.879722° E, and 617 m, respectively) was a

low-resistance zone with a width of approximately 180 m and a depth of greater than 130 m. After the 56th electrode

measurement point, the uppermost layer was a low-resistance section, with a depth of greater than 20 m, beneath which was a

high-resistance zone. The measurement results are shown in Fig. 15. The ground is rich in water at the intersection of the 75th

electrode measurement point of L1 with the 59th electrode measurement point of L2.

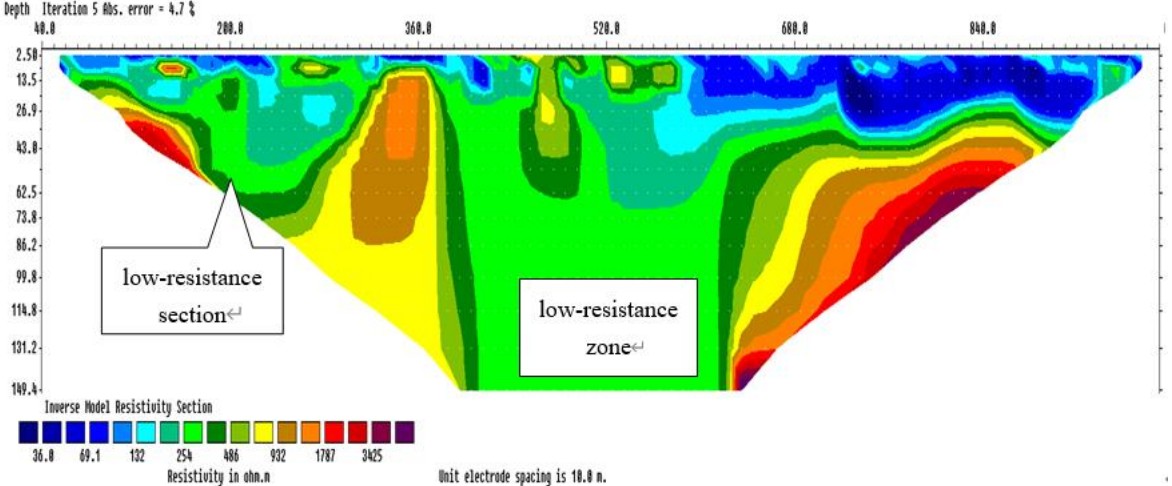

**Figure 15: Profile view of Survey Line L1 obtained using the high-density electrical method.**

**4.2.2 Survey Line L2**

Survey Line L2 had 64 electrodes, a trace spacing of 10 m, and a profile length of 640 m. The hill is more than 40 m higher

than the flat ground. The first electrode measurement point was located on the hill platform, and its latitude, longitude, and

elevation were 25.124000° N, 117.883417° E, and 663 m, respectively. There were 4 electrode measurement points on the hill

platform and 13 electrode measurement points on the slope. The last electrode measurement point was close to the village

roadside; its latitude, longitude, and elevation were 25.126611° N, 117.878528° E, and 624 m, respectively.

According to the profile view obtained using the high-density electrical method (Fig. 16), the highest part of the hill is relatively

flat, where the underground medium is relatively uniform and dense. At the foot of the hill is a low-resistance zone containing

a larger amount of underground water; the western tail of the survey line passes through the water-bearing region.





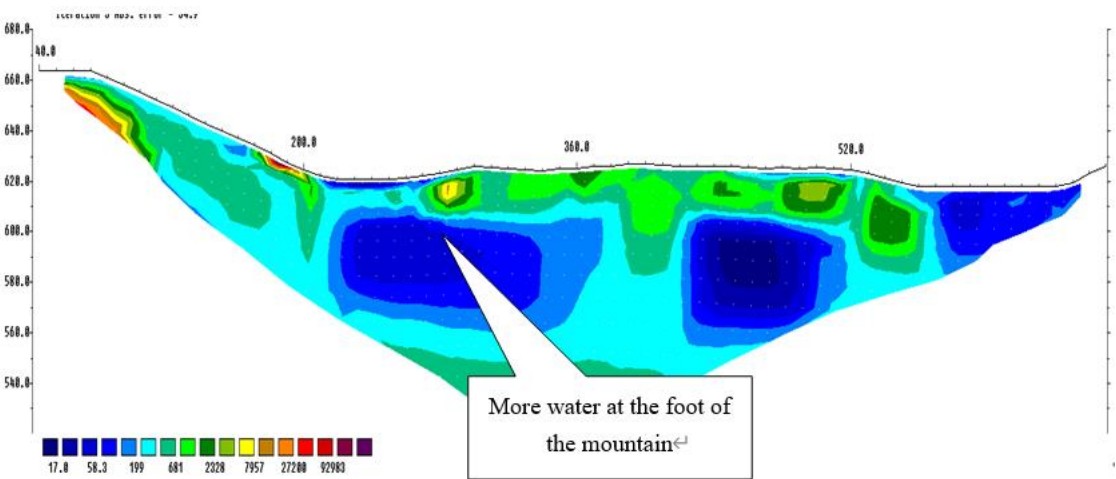

**Figure 16: Profile view of Survey Line L2 obtained using the high-density electrical method.**

### 4.2.3 Survey Line L3

Survey Line L3 was east of L1 and closer to the hill. It had 80 electrodes, a trace spacing of 10 m, and a profile length of 800 m. The latitude, longitude, and elevation of the first electrode were 25.132250° N, 117.880250° E, and 614 m, respectively. Taking the 15th electrode measurement point (with latitude, longitude, and elevation of 25.131250° N, 117.880000° E, and

613 m, respectively) as the center, the zone beneath exhibited low resistance, which corresponds to the low-resistance zone on L1 but is narrower. From the 48th electrode measurement point (with latitude, longitude, and elevation of 25.128250° N, 117.880306° E, and 616 m, respectively) to the 58th electrode measurement point (with latitude, longitude, and elevation of 25.127417° N, 117.880028° E, and 617 m, respectively) was a low-resistance zone with a width of approximately 90 m. The uppermost layer after the 58th electrode measurement point was a low-resistance section, beneath which was a high-resistance

zone, similar to the tail of L1. The profile view of L3 obtained using the high-density electrical method is shown in Fig. 17. The intersection of the 72nd electrode of L3 with the 57th electrode of L2 is rich in underground water.



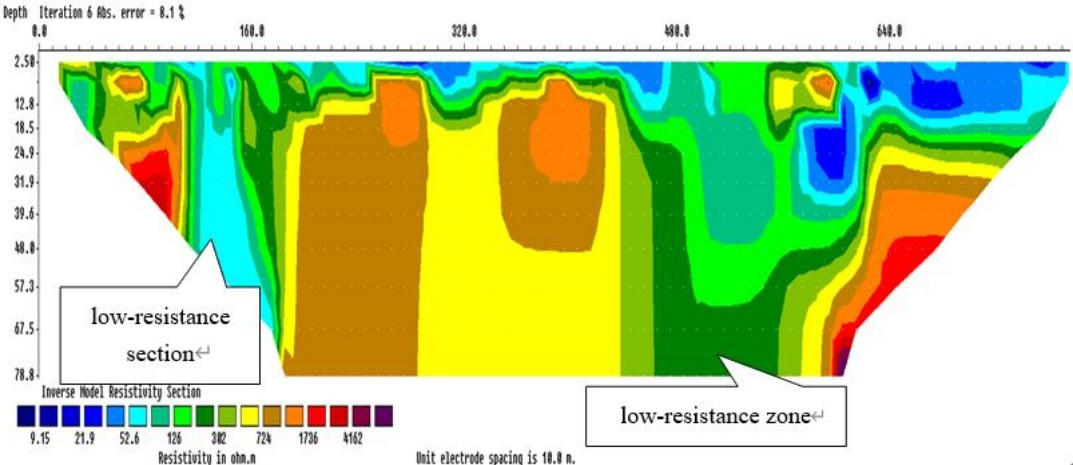

**Figure 17: Profile view of Survey Line L3 obtained using the high-density electrical method.**

**4.2.4 Discussion of the field test results**

From the profile of L2 obtained using the high-density electrical method, it can be seen that the terrain on the hill is relatively flat, where the underground medium is relatively uniform and dense, and that there are no faults, karst caves, or abnormal bodies in the surrounding area, making it a suitable site for construction of a geoelectric observatory station. From the low-resistance zone in the electrical method profile of the three survey lines, it can be inferred that this area is likely to have been

an ancient river course, which was filled in as a result of geological changes and erosion.

**5 Conclusions**

This paper has described the development of an IoT-based 4D high-density electrical instrument system; the hardware and software design of the electrical instrument and the system's performance indicators are also briefly introduced. By combining the system with cables, electrodes, and electrode converters, tests were carried out in the laboratory, on campus, and in the

field. The acquired data were processed and analyzed, and the results show that the system can be applied to perform actual field measurements.

The current trend in the development of electrical instruments is toward the increased use of multiple channels, multiple parameters, multiple dimensions, multiple functions, high power, and wide ranges (Yan et al., 2012). Therefore, our research group plans to continue improving the instrument, for example by upgrading the channel board to a multi-channel version,

increasing the power of the transmitting board, and adding a GPS module.



**Data availability.** There are no publicly available data for this study.

**Author contribution.** The author worked as the hardware design and post-debugging throughout the development process, and performed the tests, as well as the drafting of the manuscript.

**Competing interests.** The authors declare that they have no conflict of interest.

**Acknowledgments**

This work was supported by the National Natural Science Foundation of China (No. 42074155 and No. 41574131), the PetroChina Innovation Foundation (No. 2019D-5007-0302), and the Fundamental Research Funds for the Central Universities of China.

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
