# Peer review of "IoT-based 4-dimensional high-density electrical resistivity instrument for geophysical prospecting"

_Geoscientific Instrumentation, Methods and Data Systems, 2021_

## Author Response (AR1)

**Reviewer #1**

**1. The references need to be enriched;**

**Answer:** Some references have been added:

He, C. L., Shen, M. X., Liu, L. S., Yang, J., and Shi, H.: Design and realization of a greenhouse temperature intelligent control system based on NB-IoT, J. South China Agricultural University, 2, 117–124, 2018.

Li, G. G., Li, X. W., and Wen, X. C.: Influence of Internet of Things technology on the development of automatic environmental monitoring system, Environmental Monitoring in China, 1, 5–10, 2011.

**2. Wireless communication technology is used between a control center and a high-density electrical instrument, how far does it support transmission?**

**Answer:** Through 4G Internet, there is no distance limit to connect to the server. The serial port connects with your device and transfers data to your server. As long as you can stay in the network, unlimited transmission distance. Therefore, as long as there is a base station nearby, it can be operated anywhere with a network.

**3. Complete AB front-end resistance channel in Figure4;**

**Answer:** The content in figure 4 has been modified according to your suggestions.

[Figure]

**Figure 1: Block diagram of the AB.**

**4. Complete the remote interface of Figure 12 and provide the details of the remote interface;**

**Answer:** The content in figure 12 has been modified according to your suggestions.

[Figure]

**Figure 2: Real-time profile view of the test site as displayed on the remote PC.**

5. **I recommend enhancing the Conclusions by mentioning limitations of the instrument developed in your study or potential barriers to its use.**

**Answer:** Limitations of the instrument have been added:

The current trend in the development of electrical instruments is toward the increased use of multiple channels, multiple parameters, multiple dimensions, multiple functions, high power, and wide ranges (Yan et al., 2012). From this point of view, although our instrument successfully addresses a number of shortcomings of high-density electrical instruments currently available on the market for use in geophysical prospecting, including bulkiness, weight, limitations in data acquisition accuracy, and difficulty of connecting to the Internet for remote monitoring, there are still some aspects that need to be improved. For instance, it only has one channel and it does not have a GPS module. Therefore, our research group plans to continue improving the instrument, for example by upgrading the channel board to a multi-channel version, increasing the power of the transmitting board, and adding a GPS module.

**Reviewer #2**

1. **Figure 13 do not indicate the position of three survey lines clearly, replace Figure 13 with a map or contour map.**

**Answer:** The content in figure 13 has been modified according to your suggestions.

[Figure]

**Figure 3: Schematic layout of the survey lines. (The 75th measurement point on the L1 survey line intersects with the 59th measurement point on the L2 survey line, and the 72nd measurement point on the L3 survey line intersects with the 57th measurement point on the L2 survey line.)**

2. **A 4G module is employed to provide a real-time remote control and data acquisition monitoring system based on the cloud platform, can the instrument save the data itself?**

   **Answer:** The instrument can save the data as rtf and txt files because a SD card is designed in this instrument.

3. **The instrument developed in your study has optimized the high-density electrical instruments currently available on the market, but does it have any limitations or shortcomings?**

   **Answer:** The instrument developed in this study has optimized the high-density electrical instruments currently available on the market, but it still has some shortcomings, for example, it does not have a GPS module and there is only one channel on the channel board. We still need to continue improving it.

   Sincerely appreciate for reviewing the manuscript in your busy schedule.

---

## Author Response (AR2)

Sincerely appreciate for reviewing the manuscript in your busy schedule.

**Reviewer #1**

1. **The references need to be enriched;**

   **Answer:** Some references have been added:

2. **Wireless communication technology is used between a control center and a high-density electrical instrument, how far does it support transmission?**

   **Answer:** Through 4G Internet, there is no distance limit to connect to the server. The serial port connects with your device and transfers data to your server. As long as you can stay in the network, unlimited transmission distance. Therefore, as long as there is a base station nearby, it can be operated anywhere with a network.

3. **Complete AB front-end resistance channel in Figure4;**

   **Answer:**    The content in figure 4 has been modified according to your suggestions.

4. **Complete the remote interface of Figure 12 and provide the details of the remote interface;**

   **Answer:** The content in figure 12 has been modified according to your suggestions.

5. **I recommend enhancing the Conclusions by mentioning limitations of the instrument developed in your study or potential barriers to its use.**

   **Answer:** Limitations of the instrument have been added:

**Reviewer #2**

1. **Figure 13 do not indicate the position of three survey lines clearly, replace Figure 13 with a map or contour map.**

   **Answer:** The content in figure 13 has been modified according to your suggestions.

2. **A 4G module is employed to provide a real-time remote control and data acquisition monitoring system based on the cloud platform, can the instrument save the data itself?**

   **Answer:** The instrument can save the data as rtf and txt files because a SD card is designed in this instrument.

3. **The instrument developed in your study has optimized the high-density electrical instruments currently available on the market, but does it have any limitations or shortcomings?**

**Answer:** The instrument developed in this study has optimized the high-density electrical instruments currently available on the market, but it still has some shortcomings, for example, it does not have a GPS module and there is only one channel on the channel board. We still need to continue improving it.

I also improved resolution of these figures, and Fig. 9 has been modified: horizontal axis label is the number of tests. Apart from that, I added boxes/arrows to indicate the various system components clearly to Fig 10 and Fig.14. For Fig. 11 and Fig. 12, I added boxes/arrows near all buttons and text to translate Chinese text for reader's comfort. I improved the quality of Fig. 15, 16, 17. Not only can high density resistivity method be used in this instrument but also other method can be used in this instrument, like induced polarization (IP) method, which is also widely used in geophysical prospecting.

Sincerely appreciate for reviewing the manuscript in your busy schedule.